# Flow Pattern and Erosion in a 90-Degrees Sharp Bend around a W−Weir

**Vida Atashi** [1,*] **, Mahmood Shafai Bejestan** [2] **and Yeo Howe Lim** [1]

1   Department of Civil Engineering, University of North Dakota, Grand Forks, ND 58202, USA
2   Faculty of Water and Environmental Engineering, Shahid Chamran University of Ahvaz,
    Ahvaz 61357-43311, Iran
*   Correspondence: vida.atashi@und.edu; Tel.: +1-7018852449

**Abstract:** Different flow-altering methods, such as W−Weirs, have been developed to reduce erosion. For this study, we performed two experiments: (1) installing a W−Weir in various positions to determine the best angle for placement, and (2) investigating the variation of flow patterns and bed shear stress distribution in a 90-degree sharp bend by measuring the 3D components of flow velocities, with and without W−Weirs, where the greatest scour depth occurs. The results from the three installation angles indicate that less scour depth and volume of sediment removal occur when the weir is located close to the end of a bend. In addition, the value of the secondary circular power without a weir was higher than the position with a weir; however, this value significantly increased at 70 degrees due to turbulence flow near the W−Weir. This secondary flow power reduction at 45 degrees with a W−Weir increased by 65.8 percent for a Froude number value of 0.17, and by 29.8 percent for a Froude number value of 0.28, compared to values without the W−Weir, respectively.

**Keywords:** W−Weir; 90-degree bend; secondary flow; shear stress; erosion; flow pattern; Froude number

## 1. Introduction

A river eddy is a region of turbulent water that develops behind an obstacle. Often, the water in the eddy will reverse the direction of flow and will flow upstream. The eddy flow in the river bend forces the velocity vectors towards the surface's external bend and into the bed's inner bend. Eddy flows are caused by changes and transformations: the maximum flow velocity in the bend is close to the cut bank of the riverbed, leading to an increase in shear stress and erosion. The primary characteristic of flow in river bends is the creation of eddy flows from secondary and main flow interactions. Secondary flow occurs when the two pressure gradients and centrifuge forces interact [1], and are caused by turbulence anisotropy (Prandtl's secondary flows of the second kind) [2]. Secondary currents at the water's surface move low-momentum fluid from the side walls to the middle of the channel, while high-momentum fluid is inhibited below the free surface.

Cross-movement due to secondary flow is observed in the channel bed. Erosion, sedimentation, and changes in river morphology are critical variables to consider when evaluating communication routes, riverbank facilities, and installations. Establishing appropriate locations for water intakes, evaluating bank and bed stability, and diverting riverbed flow have been topics researched for boating and locating facilities which are used to transfer pollutants into rivers. Eddy flow has a significant role in the creation of a cross-section profile and shear stress in the bed.

Leschziner (1979) studied the flow pattern in a sharp bend, and observed that the place of maximum velocity tended to occur towards the external wall as it approached the end of the bend [3]. He noted that the longitudinal pressure gradient was the primary factor driving the maximum velocity towards the external wall, while the maximum velocity of secondary flow was the main factor of dislocation in mild bends. Odgaard and Bergs (1988) introduced a simple analytical model to describe the specifications of flow and shear stress

in a bend's alluvial channels with a non-homogenous shear stress distribution. That model described the relationship between secondary flow and the cross changes of shear stress. The presented model simulated the behavior of measured flow in laboratory channels using latitudinal variations of the artificially made shear stress [4]. Lee et al. (2019) concluded that the maximum shear stress occurred at the beginning of the bend near the inner wall and at the end of the bend near the outer wall, for a radius of curvature of less than three [5]. The results indicated that the erosion of the outer wall in low flow conditions, with a Froude number of 0.35, began upstream from the top of the bend. It then moved to the upstream and downstream sides of the bend, with greater depth at medium and high flow conditions, and with Froude number values of 0.39 and 0.43. Ghadoo and Shafai (2014) measured three-dimensional velocity in a sharp bend under different channel wall roughnesses, calculated the shear stress distribution of the bed, and compared the results to other studies. The authors concluded that some areas were favorable for bed erosion and sedimentation next to the internal and external walls in a 90-degree sharp bend, from the 70-degree point to downstream [6]. Their results indicated that an increase in wall topography coarseness led to a reduction in the resultant secondary flow power and transfer of secondary flow downstream, which could be an effective factor in decision-making in a meandering river's bank protection design.

Three recent studies that specifically addressed this concern were noted in a literature review on stage-discharge relationships related to river-spanning rock weirs. Ruttenberg (2007), Meneghetti (2009), and Thornton et al. (2011) developed river-spanning rock weir stage-discharge relationships [7–9]. Ruttenberg (2007) created a stage-discharge connection for U-Weirs using measured field data from three sites along Beaver Creek in north-central Washington. He developed equations to calculate river discharge based on weir geometry, specifically the wetted weir length along the weir crest. Calculating wetted weir length, the variable we are attempting to estimate, requires weir geometry and water stage; therefore, his calculations could not be used in the design process. Thornton et al. [7] investigated the hydraulic efficiency of labyrinth-shaped rock weirs and proposed head-discharge equations based on weir length, the head upstream, weir height, rock size, and discharge.

Different methods are used to minimize the impacts caused by eddy flow in river bends and to improve and restore the edges of rivers and their beds. The cross-vane, W−Weir, and J-hook vane are examples of structures that can be used to maintain or improve river stability and functionality, as well as to accomplish various other objectives. These structures have been effectively used in natural channel design for recreational boating, irrigation diversions, fish habitat improvement, bank stabilization, grade control, and river restoration [1,10–13].

Few studies have been conducted on weirs regarding improving flow patterns and river restoration. Abdollahpour et al. (2017) investigated the patterns of erosion and sedimentation downstream of a W−Weir. They examined a sinusoidal channel with a sinuosity of 1.12. The key criteria influencing scour size and shape were the inflow Froude number, weir height, and the angle between two inner arms of the W−Weir. According to the study's findings, reducing the height of the W−Weir reduced scour volume, on average, for different input Froude numbers [14]. Kurdistani and Pagliara [15] investigated the scour phenomena and bed morphologies downstream of curved channel cross-vane structures. Pagliara et al. [16] investigated the scour phenomenon downstream of block ramps, demonstrating that reducing the curvature radius resulted in a three-dimensional equilibrium scour morphology. Empirical equations were developed based on a dimensional analysis to predict the maximum scour depth for various combinations of hydraulic conditions, channel bend, structure orientation corresponding to flow direction, and structure geometry. The primary parameters affecting scour size and morphology were the densimetric Froude number, drop height, tailwater, structure height, and channel curvature. Structure orientation did not affect scour depth and length, nor ridge height and length.

Structures are usually constructed without considering sediment transportation, violating the stabilized river's dimension, pattern, and profile [17]. Knowing precisely how

scour morphology changes under various hydraulic situations is crucial [15]. Only a limited number of studies have used W−Weirs in a meandering channel to determine how the sharp bend affects scouring.

Bhuiyan et al. (2007) conducted field studies on a large spiral-shaped water channel with movable beds [10]. The authors placed a zigzag weir immediately downstream of the bend. Their results indicated that the region with maximum transfer moved to the middle parts of the channel in the presence of a W−Weir; however, no significant changes occurred to the maximum amount of sediment transfer. The carried load in the overall discharge rate was 5 percent more when a W−Weir was used than in a normal situation. The displacement weir on the right side of the channel's central line, or central bend, exhibited a slight increase due to the impacts of erosion holes and low inversion levels. The sediments on the right hand of the channel's central line were larger and had a lower flatness coefficient after the weir was installed; however, the carried materials were smaller and had a greater flatness coefficient close to the outer bank. Sediments carried on the inner side of the channel seemed smaller and more homogeneous than those on the entire bank, except for a tape of large materials with a high flatness coefficient. Bhuiyan et al. (2009) investigated the flow turbulence characteristics and scour development downstream of the W−Weir and vanes at the river bend, both in clear water and under live-bed conditions [18]. The authors conducted their studies in sinuous channels, 1.6 m wide, with a 1.38 sinuosity level and 0.00133 longitudinal slopes. Both W−Weir arm angles and the angle at which the vanes were attached to the downstream bank were 30 degrees. The W−Weir installation caused the formation of two scour holes, and had no effect on the sediment flows downstream of the channel bed or along the meandering channel.

Atashi et al. (2016) established that a W−Weir causes the emergence of a still flow upstream of the structure, indicating that the secondary flow, which is established, either disappears entirely with W−Weir or is transferred to the center direction of the flow. The maximum dimensionless shear stress occurred in the external wall range of 20 to 40 cm. A secondary flow was observed in the external wall at a 90-degree angle after the W−Weir was installed [19].

The purpose of this research was (1) to install a W−Weir at different angles to study the topographical parameters of the bed and the scale of erosion in the inner and outer bend, as well as to identify the best location to install the W−Weir; and (2) to investigate the cross velocity, shear stress, and secondary circular power in a 90-degree sharp bend based on the best location for W−Weir installation.

## 2. Materials and Methods

### 2.1. Experiments

We have experimentally analyzed the scour geometry downstream of a W−Weir in a bent channel, as well as the flow pattern and shear stress changes in different discharges and locations along the inner and outer bends. This weir had a sloped crown and a "W" shape when viewed from downstream. The side corners of the W had 20 to 30-degree angles to the banks. The first peak was located at 25 percent of the width, the second at 50 percent, and the third at 75 percent [10]. Figure 1 depicts a W−Weir located on a river bend and its dimensions.

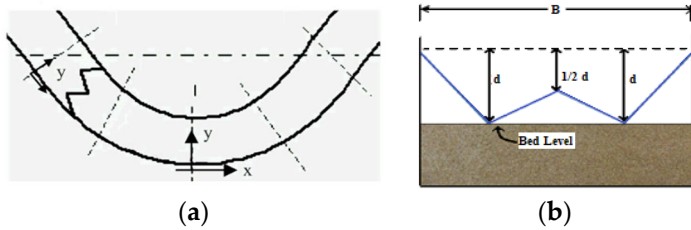

(a)                                        (b)

**Figure 1.** (a) W−Weir in the river bend and (b) the W−Weir shape [18].

A physical model was built and prepared in the hydraulic laboratory of the Shahid Chamran University, Ahwaz, Iran, to achieve this study's goals. The laboratory equipment included a suction pool, pumping station, head supply tank, auxiliary tank, measurement equipment, and discharge rate regulation tools, such as a triangle-shaped weir and a 6-inch (15.24 cm) slide valve. Raudkivi and Ettema (1983) suggested that the average diameter of the particles should be greater than 0.7 mm to prevent the creation of ripples [20]. Another concern was that there not be any scour in the direct area of the flume at high discharges. Sieve analysis in the Soil Mechanics Laboratory at the Shahid Chamran University of Ahvaz yielded an accurate grain size chart of the sediments. Following the preliminary testing, sediment with a mean diameter of $d_{50}$ = 1.5 mm was chosen as the sediment utilized in the study. In all experiments, the sediment size distribution was uniform, with a geometric standard deviation of $Sg$ = 1.92 ($Sg = \sqrt{d_{84}/d_{16}}$ , $d_{84}$ = 2.4 mm, and $d_{16}$ = 1.25 mm). According to the formula, $C_u = D_{60}/D_{10}$, the uniformity coefficient, was calculated to be 1.63. $D_{10}$ is called effective particle size. This means that 10 percent of the particles are finer, and 90 percent of the particles are coarser than $D_{10}$. Similarly, $D_{60}$ is the particle size at which 60 percent of the particles are finer and 40 percent of the particles are coarser than the $D_{60}$ size. Figure 2 depicts the plan of the laboratory flume. $R/b$ = 2, where $R$ is the radius of the bend and b is the width of the flume; therefore, the bend is categorized as a sharp bend [7].

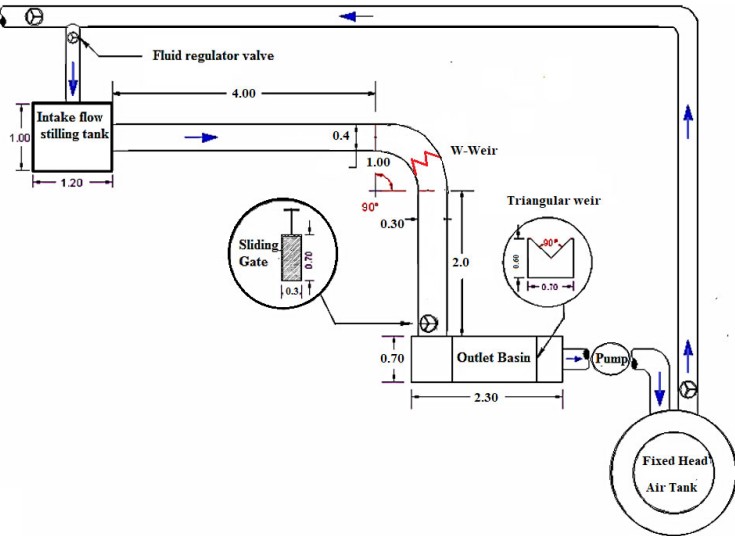

**Figure 2.** Plan view of laboratory flume and the test bend (all dimensions in meters).

A W−Weir, constructed using a 1-mm-thick galvanized sheet, was installed at different angles of the bend (Figure 3).

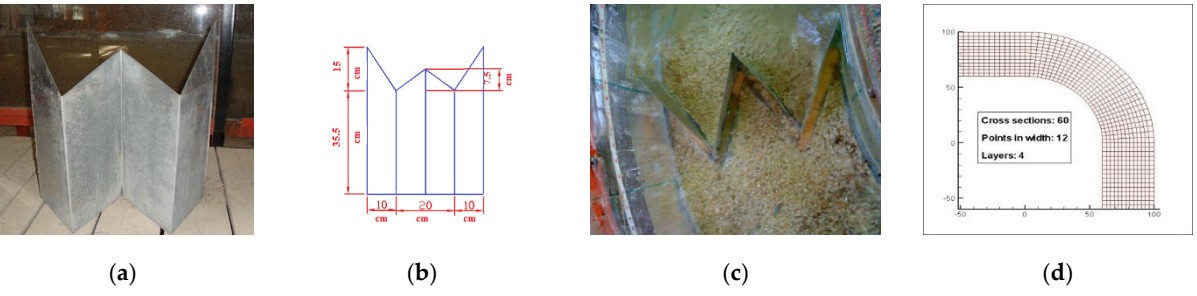

| (a) | (b) | (c) | (d) |

**Figure 3.** Picture of the W−Weir and its sizes: (**a**) W−Weir, (**b**) size of W−Weir based on the Bhuiyan et al. study [18], (**c**) W−Weir installed in the flume, and (**d**) plan view of the mesh for measurement nodes.

The test was performed as follows: the flow slowly entered the flume as the valve was gradually opened, and the water surface in the flume rose. The downstream gate was then

progressively opened to prevent the water surface from lowering below a specified level. A triangle weir with a sharp edge also measured the amount of flow passing the flume at the same time. The operations proceeded until the appropriate discharge rate was determined. The channel wall was made of plexiglass to allow for visual observations. The tests were performed at discharge rates of 15 and 25 L/s, with Froude number values of 0.17 and 0.28 and a constant flow depth of 17 cm. The inflow discharge rate and water surface level were controlled constantly throughout each test's four- to eight-hour duration. The flow depth, flume height, and layer arrangement were fixed at 17 cm to accurately compare the results and the conformity of the Froude numbers. A W−Weir was installed at three different angles, 30-, 60-, and 90-degrees, for the first set of experiments to determine the best angle for weir installation. Data collection began after 90 min of testing, and once the flume was emptied, a laser meter was used to obtain the distance from the transmission source to the sediment surface. Data were collected in the direction of the 90-degree bend of the sections perpendicular to the flume wall at 2.5 degrees intervals, and the straight direction downstream of the bend every 10 cm, to improve accuracy.

Velocity measurements began once the best angle for W−Weir installation was chosen, and the required hydraulic conditions were achieved. A JFE ALEC velocity meter using 20 Hz frequency was utilized to measure the three-dimensional components of flow velocity at various flow discharge rates.

After determining the best placement for the W−Weir, the three categories of channel bend velocities were divided into five cross-sections at zero-, 45-, 70-, and 90-degree angles to determine the measurement nodes, with 12 orthogonal lines used for each cross-section. Each orthogonal line was divided into four layers, 3, 7, 11, and 15 cm from the bed. Measurements were also taken directly upstream and downstream of the flume at 90 degrees and a 30 cm distance from the zero points (Figure 3d). In this way, 336 nods were obtained in each test for measuring velocity categories. Table 1 shows the scenario of the tests that were performed in this research.

**Table 1.** Test programs of this research based on flow conditions.

| Discharge Rate (L/s) | Flow Depth (cm) | Froude Number | Points Measured (Degree) |
|---|---|---|---|
| 15<br>35 | 17 | 0.17<br>0.40 | 0<br>45<br>70<br>90 |

## 2.2. Secondary Circular Power

In this study, the secondary circular power of the flow in the bend was studied with and without W−Weir to obtain a more precise observation of the impact of W−Weir installation. The secondary circular power was one of the methods used to calculate the secondary flow power. Equation (1) was used to calculate the secondary circular power of the secondary flow. Velocity is lowest along the bed and walls of the channel because the water encounters more resistance to the flow. The maximum velocity along a straight length is observed near the surface in the midchannel. The highest velocity of the water swings to the outside of the channel whenever it rounds a bend. If secondary flow lines are assumed to be a plain, in a way that the difference in the velocity of both sides of the plain shows power, it will be possible to use equation 1 for calculating the secondary flow power [21].

$$\delta = U_{\max} - U_{\min} \tag{1}$$

where $\delta$, $U_{\max}$, and $U_{\min}$ are the flow secondary circular power and maximum and minimum velocities, respectively, in meters per second.

### 2.3. Shear Stress of the Bed

It is critical to determine the shear stress distribution in meandering channels to predict bank erosion, since it is a significant factor in sediment transport mechanisms. The Reynolds shear stress method [22], Preston tube method [23,24], linear regression method [22,25], and depth-averaged method [26] are a few of the many methods used to characterize shear stress distribution. The depth-averaged method was used to calculate the bed shear stress distribution performed in polar form $p(r,\theta)$. The following relations were used to change the polar coordinates to cartesian coordinates:

$$U = U_\theta \cos\theta + U_r \sin\theta \tag{2}$$

$$r = r' + \Delta r, \ \ x = r\sin\theta, \ \ y = r\cos\theta \tag{3}$$

$$V = U_\theta \sin\theta - U_r \cos\theta \tag{4}$$

where $\Delta r$ is the distance to the measurement point from the internal wall (centimeter), $U_r$ and $U_\theta$ are the bi-dimensional velocities in polar coordinates (cm/s), $U$ and $V$ are the bi-dimensional velocities in cartesian coordinates (cm/s), and $R$ is the radius of the internal bend, which is equal to 60 cm.

The shear stress of beds in meandering channels has been extensively investigated [27–30]. Bathurst et al. (1979) proposed Equations (5) and (6) to determine bed shear stress in the $x$ and $y$ directions [31]. The Chézy coefficient is the resistance factor, and establishes the hydrodynamic behavior of the flow bed, which is calculated from $c = 1/n \ R^{1/6}$ relations, where $R$ is the hydraulic rate radius of the flow cross-section in meters and n is the Manning coarseness coefficient [32]. Manning's relation is one of the most important relationships proposed to determine the value of the Chézy coefficient.

$$\tau_{bx} = \frac{\rho g}{c^2} \bar{U} \sqrt{\bar{U}^2 + \bar{V}^2} \tag{5}$$

$$\tau_{by} = \frac{\rho g}{c^2} \bar{V} \sqrt{\bar{U}^2 + \bar{V}^2} \tag{6}$$

where $\bar{U}$ and $\bar{V}$ are the average depth of velocity in the $x$ and $y$ directions (m/s), $\tau_{bx}$ and $\tau_{by}$ are the shear stresses of the bed in the $x$ and $y$ directions (N/m$^2$), and $\rho$ and g are the fluid density and gravity coefficient, respectively.

## 3. Results and Discussion

### 3.1. Topography and Scour for Different W−Weir Installation Angles

Figures 4 and 5 depict the results of tests performed with the presence of the W−Weir after 90 min of relative equilibrium time. These images demonstrate the bed's topography with Froude number values of 0.17 and 0.28 and W−Weir installations at 30-, 60-, and 90-degree angles. The beginning of the bend is marked at zero degrees, and the end of the bend is marked at 90 degrees.

The scour depth values with a Froude number of 0.17 are marked on the horizontal lines and by the color spectrum (Figure 4a). The maximum depth of erosion was 0.064 m, which occurred at the location of the 30-degree angle, where the distance from the outer wall was zero centimeters, and there were almost zero topographical changes upstream from the structure. The maximum depth of erosion was equal to 0.062 m (Figure 4b), which occurred at the 50-degree angle of the bend, where the distance from the outer wall was 10 cm. The erosion was almost zero upstream (Figure 4c). The scour depth values are marked on the horizontal lines and by the color spectrum. The maximum depth of erosion was equal to 0.039 m at an angle of 82.5 degrees of the 90-degree bend, where the distance from the outer wall was 7 cm and the topography changes were almost zero upstream from the structure.

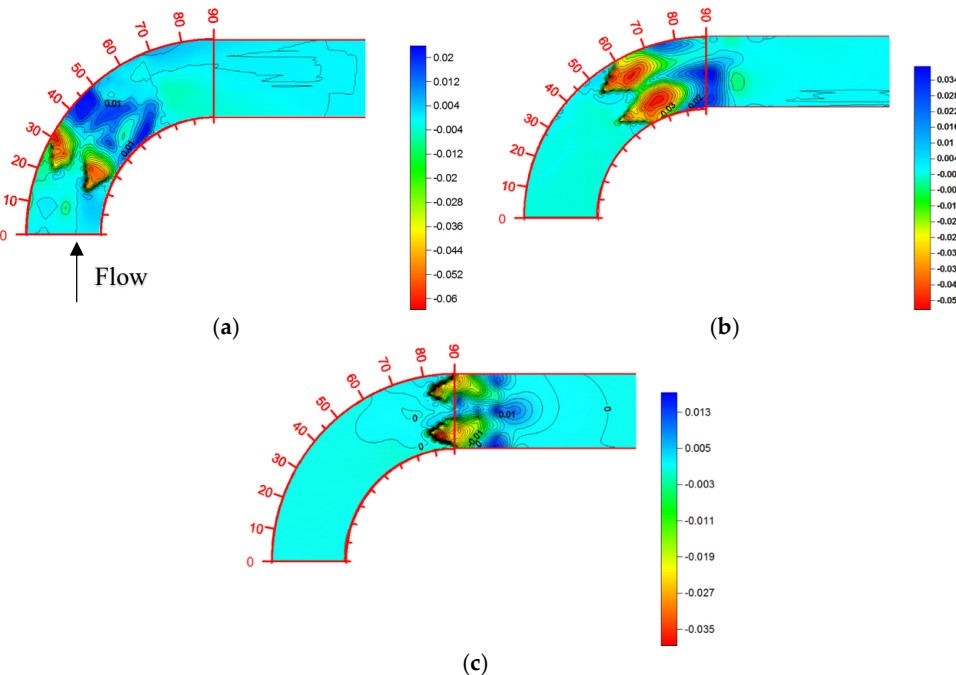

**Figure 4.** The topography of the bed for a Froude number of 0.17 and W−Weir installation at 30, 60, and 90 degrees (all dimensions in meters). (**a**) Installation angle = 30 degrees. (**b**) Installation angle = 60 degrees. (**c**) Installation angle = 90 degrees.

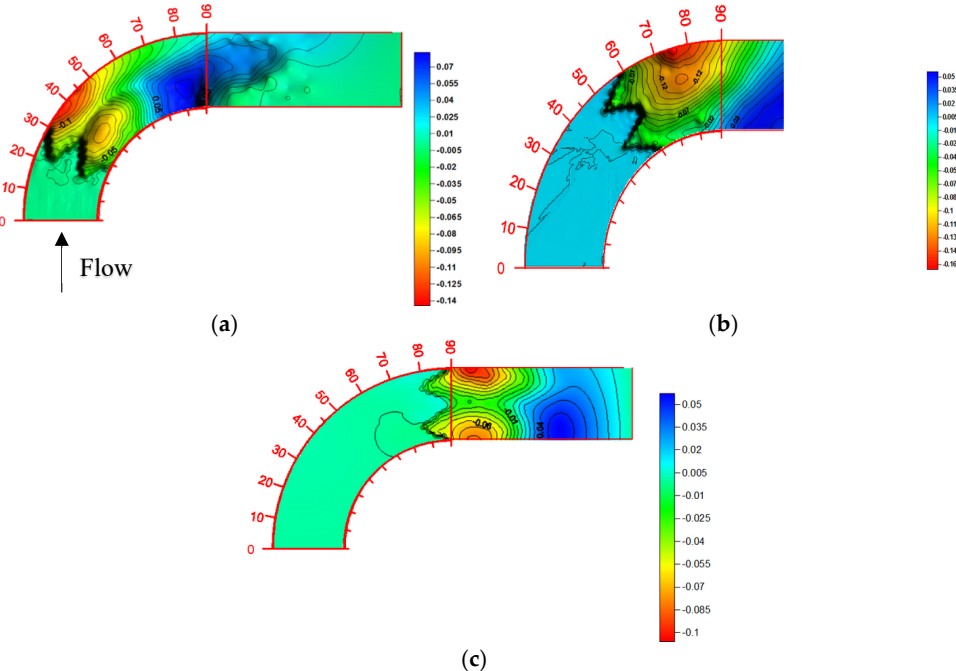

**Figure 5.** The topography of the bed for a Froude number value of 0.28 and W−Weir installation at 30, 60, and 90 degrees (all dimensions in meters). (**a**) Installation angle = 30 degrees. (**b**) Installation angle = 60 degrees. (**c**) Installation angle = 90 degrees.

The maximum scour depth values for a Froude number of 0.28 are indicated with horizontal lines and a color spectrum in Figure 5a. The maximum depth of erosion was equal to 0.147 m, which occurred at the 45-degree angle mark of the 90-degree bend, where the distance from the outer wall was zero centimeters. The topography changes above the structure were close to zero. The maximum scour was equal to 0.166 m, which occurred at

75 degrees, where the distance was 2 cm from the outer wall (Figure 5b). The topography changes upstream of the weir were almost zero. The maximum scour depth in Figure 5c was equal to 0.107 m, which occurred at 10 cm of the straight path downstream of the 90-degree bend at the zero point of the outer wall. There was almost no erosion above the weir. Table 2 provides a summary of the experiments described above.

**Table 2.** The maximum depth of the scour holes for different Froude numbers and W−Weir installation at 30, 60, and 90 degrees.

| Froude Number | W−Weir Installation Angle (Degrees) | Scour Angle (Degrees) | Maximum Scour Depth, ΔZ (m) | Profile |
|---|---|---|---|---|
| | 30 | 45 | 0.147 | Outer bend |
| | | 40 | 0.083 | Centerline |
| | | 40 | 0.094 | Inner bend |
| 0.17 | 60 | 75 | 0.166 | Outer bend |
| | | 75 | 0.125 | Centerline |
| | | 77 | 0.097 | Inner bend |
| | 90 | 10 cm after the bend | 0.107 | Outer bend |
| | | 10 cm after the bend | 0.020 | Centerline |
| | | 10 cm after the bend | 0.085 | Inner bend |
| | 30 | 30 | 0.064 | Outer bend |
| | | 30 | 0.014 | Centerline |
| | | 25 | 0.051 | Inner bend |
| 0.28 | 60 | 50 | 0.062 | Outer bend |
| | | 60 | 0.012 | Centerline |
| | | 65 | 0.051 | Inner bend |
| | 90 | 82.5 | 0.039 | Outer bend |
| | | 75 | 0.001 | Centerline |
| | | 82.5 | 0.039 | Inner bend |

Figure 6 depicts the erosion depth changes in the outer part of the bend at 30, 60, and 90 degrees for Froude number values of 0.17 and 0.28. The highest amount of erosion was in the outer wall, and the most bed profile fluctuation was at the installation angle of 30 degrees, for a Froude number of 0.17. The lowest erosion depth was at 90 degrees (Figure 6a). The highest erosion depth was at the outer part of the bend, the highest bed profile fluctuation was at the installation angle of 60 degrees, and the lowest was at 90 degrees for a Froude number of 0.28 (Figure 6b). An increase in Froude number, from 0.17 to 0.28, caused a deeper hole in the same W−Weir installation.

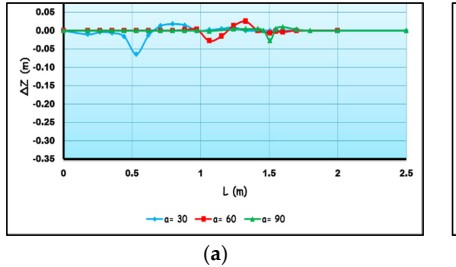 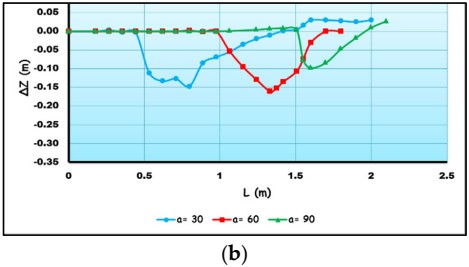

(**a**) (**b**)

**Figure 6.** Comparison of erosion depth changes in the outer wall of a 90-degree bend for the Froude number values of 0.17 and 0.28 and W−Weir installation at 30, 60, and 90 degrees. (**a**) Froude number = 0.17. (**b**) Froude number = 0.28.

The depth of the erosion hole in the outer wall with a 90-degree installation angle decreased by 27.2 percent and 35.5 percent, compared to a 30-degree and 60-degree angle of installation, respectively, with a Froude number of 0.17. The depth of the erosion hole in the outer wall decreased by 39 percent and 37 percent with a Froude number of 0.28 and a 90-degree installation angle compared to the 30- and 60-degree installation angles,

respectively. Installing the W−Weir at a 90-degree angle yielded better performance in controlling the erosion of the outer side and the topographic changes based on observations.

Ghadoo and Shafai (2014) established that the maximum amount of flow turbulence occurs at an angle of 70 degrees during weir installation [6]. Bhuiyan (2009) determined that the weir should be installed immediately downstream of the curve, where the flow pattern has severe turbulence [18]. Atashi et al. (2016) [19] revealed that the maximum depth and volume of sediment removal of erosion reduced as the W−Weir installation point moved closer to the end of the bend; therefore, a 70-degree angle was chosen as the optimal angle for the weir installation. It should be noted that the current study's experimental setup is the same as that of Atashi et al. (2016) and Ghadoo and Shafai (2014). The second half of the bend was proposed by Atashi et al. (2016), and 70 degrees of the bend was suggested by Ghadoo and Shafai (2014) as the ideal location to install a W−Weir. Given the similarities between these two studies, 70 degrees of the bend was determined to be the most crucial angle for the W−Weir placement. Based on this fact, flow pattern, secondary circular power, and shear stress distribution are investigated in the following sections, with and without the presence of a W−Weir at a 70-degree angle.

### 3.2. Flow Pattern

We measured three-dimensional velocity components, with and without the presence of W−Weirs, to study changes in the flow patterns. We also analyzed the effects of a W−Weir on the bed shear stress distribution and secondary power flow.

The cross-distribution of the velocity in various points for two Froude numbers, 0.17 and 0.28, with and without a W−Weir, were drawn to illustrate flow pattern changes (Figures 7–10). The X-axis is the width of the flume in centimeters, and the Y-axis is the flow depth in centimeters. Figure 7 displays the cross-velocity distribution in a zero-degree weir installation for a Froude number of 0.17, with and without a W−Weir. The effects of a W−Weir on the bed shear stress distribution and secondary power flow were analyzed.

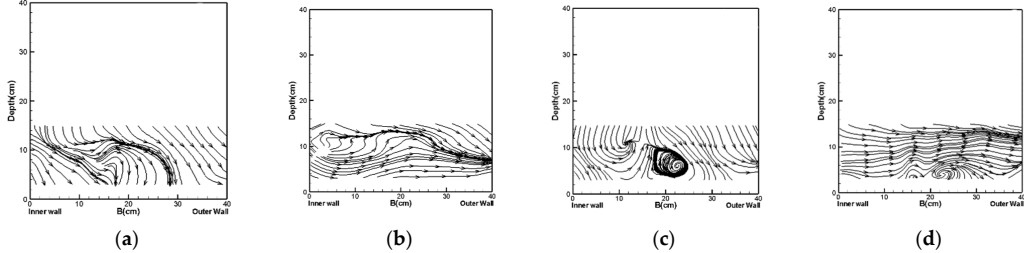

|   (a)   |   (b)   |   (c)   |   (d)   |

**Figure 7.** The latitudinal distribution of velocity at zero degrees with and without a W−Weir, (**a**) Fr = 0.17 with a W−Weir, (**b**) Fr = 0.17 without a W−Weir, (**c**) Fr = 0.28 with a W−Weir, and (**d**) Fr = 0.28 without a W−Weir.

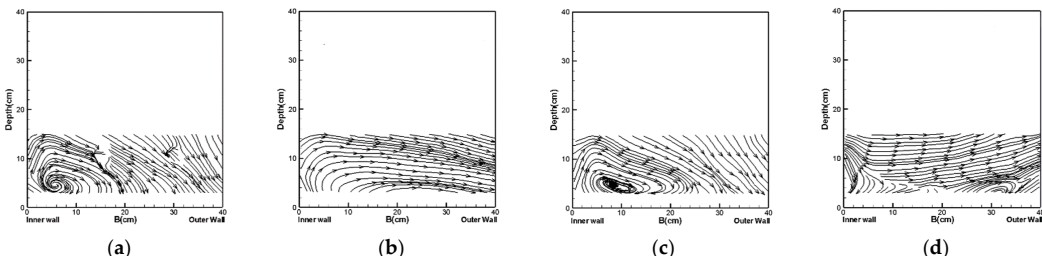

|   (a)   |   (b)   |   (c)   |   (d)   |

**Figure 8.** The latitudinal distribution of velocity at 45 degrees with and without a W−Weir, (**a**) Fr = 0.17 with a W−Weir, (**b**) Fr = 0.17 without a W−Weir, (**c**) Fr = 0.28 with a W−Weir, and (**d**) Fr = 0.28 without a W−Weir.

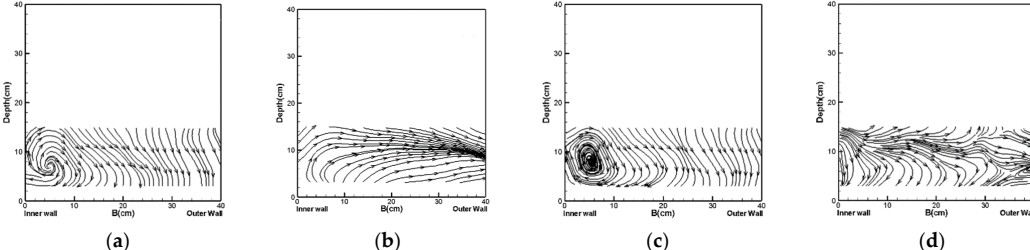

**Figure 9.** The latitudinal distribution of velocity at 70 degrees with and without a W−Weir, (**a**) Fr = 0.17 with a W−Weir, (**b**) Fr = 0.17 without a W−Weir, (**c**) Fr = 0.28 with a W−Weir, and (**d**) Fr = 0.28 without a W−Weir.

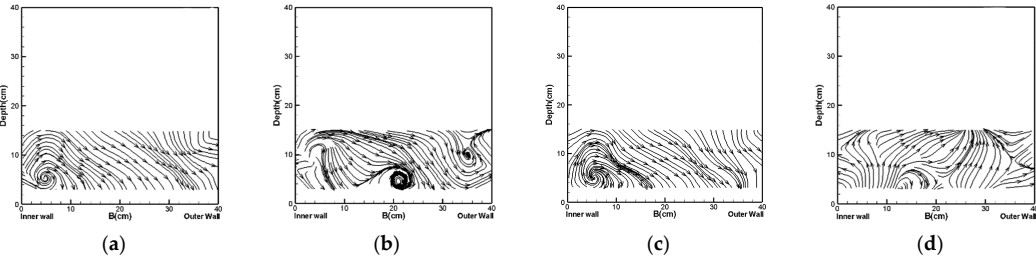

**Figure 10.** The latitudinal distribution of velocity at 90 degrees with and without a W−Weir, (**a**) Fr = 0.17 with a W−Weir (**b**) Fr = 0.17 without a W−Weir, (**c**) Fr = 0.28 with a W−Weir, and (**d**) Fr = 0.28 without a W−Weir.

The absence or presence of a W−Weir in a zero-degree position did not make noticeable changes in the flow line patterns, with an increase in the Froude number from 0.17 to 0.28. The flow has not yet entered at this point, and the behavior of the flow lines is close to the performance and behavior in direct routes. The non-entry of flow in the bend stopped the establishment of secondary flow. The presence of a weir makes the flow lines smoother for both Froude numbers (Figure 7). Figure 8 illustrates the cross-velocity distribution in a 45-degree weir installation for a Froude number of 0.17, with and without the presence of a W−Weir.

The secondary flow formed near the inner bend as the flow entered the bend (Figure 8a,c). This secondary flow is towards the outer bend on the flow surface and the inner bend near the bed. Changes in the Froude number had little effect on the secondary flow in the flow line pattern without the W−Weir (Figure 8b,d). There was more turbulent flow in the bed with the presence of a W−Weir and a Froude number of 0.28. The impacts of secondary flow formation are visible in the current center at low depth (Figure 8a,c). Figure 9 depicts the cross-velocity distribution at the 70-degree position for the two flow conditions, with or without the presence of a W−Weir.

Results for a location without a W−Weir at 70 degrees indicated that an increase in Froude number caused more intensity and secondary flow expansion as it moved toward the center line of the channel (Figure 9b,d). The changes in the Froude number in a location with a weir caused changes in the direction of the flow lines near an outer wall (Figure 9a,c). The secondary flow expanded toward the inner walls, and the surface velocity vector extended to the outer bend, while the velocity vectors extended to the inner bend at the 70-degree position due to its location in the second half of the bend. The flow lines converged and the patterns smoothed, with a Froude number of 0.17; however, the lines exhibited more turbulence with a Froude number of 0.28, and the shape of the flow lines was typically irregular (Figure 9a,c). Figure 10 depicts the cross-velocity distribution of velocity and flow patterns in a 90-degree position, with and without a weir.

The flow maintained its vortex structure at a 90-degree angle as the bend ended, even though the flow was at its end, and it followed a completely straight route. This phenomenon occurred because the flow was experiencing smoother conditions than at a

70-degree position (Figure 10). The presence of a W−Weir caused an eddy current from the internal bend to the channel's central axis, limiting erosion and other damages; however, the flow lines in the presence of a W−Weir did not follow a particular pattern, similarly to the 70-degree position. A secondary flow formed in the bed and the center of the bend for the same reason as the 70-degree case. The geometry of the current line was unaffected by the changes in Froude number, regardless of the presence of a W−Weir, for the same reason mentioned in the 70-degree situation: a secondary flow forming in the bed and center of the bend. Changes in Froude number, with and without a W−Weir, generally did not cause notable changes in the shape of the current lines.

*3.3. Secondary Circular Power*

Figure 11a,b compare the secondary flow power at various points of the bend from 0 to 90 degrees, with and without a W−Weir, for the 0.17 and 0.28 flow values.

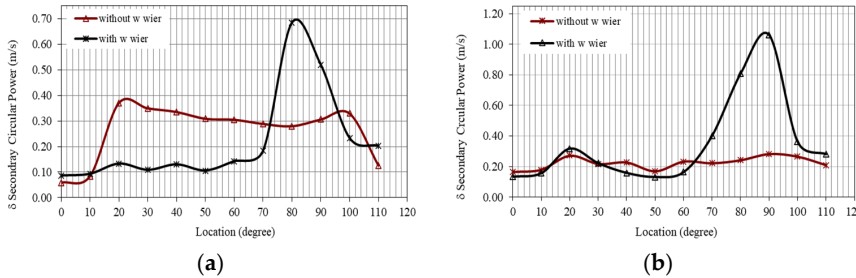

(**a**)                                                      (**b**)

**Figure 11.** Secondary circular power at different positions at the 90-degree bend with and without the presence of a W−Weir. (**a**) Froude number = 0.17. (**b**) Froude number = 0.28.

The value of the secondary circular power without a W−Weir was higher than with a W−Weir from the 10- to 70-degree position (Figure 11). With a Froude number of 0.28, the difference between secondary circular power, with and without a W−Weir, was low before 70 degrees. It should be noted that in a 20-degree position with the W−Weir, due to the full development of secondary flow, its secondary circular power considerably differs for a 0.17 Froude number, and the value of δ with the W−Weir is reduced by 64 percent in comparison to without the W−Weir. Table 3 lists the decreases and increases in secondary circular power with W−Weir installation, compared to without W−Weir installation, for the Froude number values of 0.17 and 0.28.

**Table 3.** The changes in secondary circular power of secondary flow at the associated W−Weir positions, compared to without a W−Weir, for the Froude number values of 0.17 and 0.28.

| Angle (Degrees) | Froude Number | Changes in δ of Secondary Flow in the Presence of a W−Weir Compared to without a W−Weir (Percent) |
|---|---|---|
| 0 | | −46.20 |
| 20 | | 64.25 |
| 45 | 0.17 | 65.84 |
| 70 | | 36.64 |
| 90 | | −69.94 |
| 0 | | 11.9 |
| 20 | | −2.4 |
| 45 | 0.28 | 29.8 |
| 70 | | −81.9 |
| 90 | | −275.14 |

The secondary circular power (δ) for both Froude numbers was significantly reduced in the presence of the weir from the zero-degree angle to where the weir was installed; however, this parameter, with lower Froude numbers, reduced at a greater rate (Table 3). The secondary circular power value with a W−Weir became significantly higher than without a W−Weir, because of the appearance of two eddy vortices downstream of the

structure installation. For example, the percent of reduction in the secondary flow power at a 45-degree angle with a W−Weir and 0.17 and 0.28 Froude numbers was 65.84 percent and 29.8 percent, respectively.

### 3.4. Shear Stress

As mentioned earlier, Equations (5) and (6) are proposed to determine bed shear stress in the *x* and *y* directions based on the depth-averaged method. The maximum stress of each cross-section was obtained, and is dimensionless in proportion to the average shear stress. Figure 12 depicts the shear stress distribution of the bed for the 0.17 and 0.28 Froude number values, with and without the presence of a W−Weir.

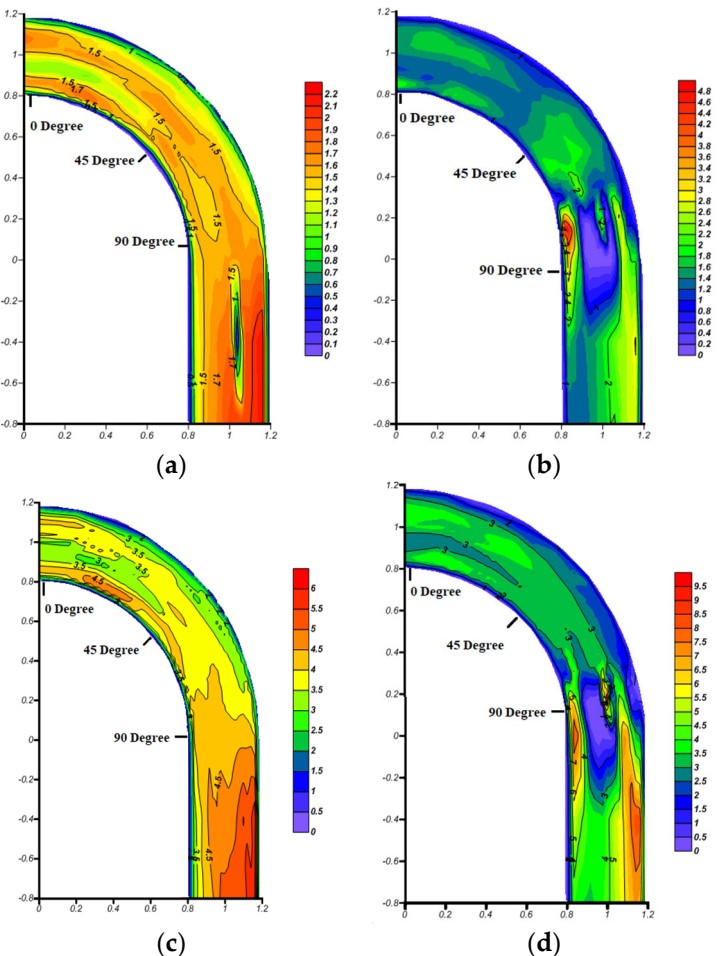

**Figure 12.** Distribution of shear stress of the bed with both states with and without a weir. (**a**) With W−Weir and 0.17 Froude number. (**b**) Without W−Weir and 0.17 Froude number. (**c**) With W−Weir and 0.28 Froude number. (**d**) Without W−Weir and 0.28 Froude number.

The presence of a W−Weir caused homogeneous shear stress before the weir installation angle, and two points with high shear stress were observed after the weir installation angle (Figure 12); therefore, the distribution of shear stress increased with an increase in Froude number. This is because an increase in Froude number causes more turbulence in the channel's current, enabling it to expand a flow with greater shear stress in a larger area.

The dimensionless shear stress values were almost constant with a Froude number of 0.17 and without the presence of a W−Weir; however, this parameter significantly increased as the Froude number increased to 0.28. Not many changes occurred in the dimensionless shear stress with a Froude number of 0.17 in various positions along the bend, while the ratio of the dimensionless shear stress experienced no significant changes with a Froude number of 0.28 at the 30-degree angle position. The value of the dimensionless shear stress

increased in value when placed at the peak of the bend and further. The presence of the weir did not have much impact on the mentioned parameters, with a Froude number of 0.17. The only difference is that the dimensionless shear stress increased after weir installation at the 70-degree point. For instance, the dimensionless shear stress increased by 2.8, 2, and 1.7 times at an angle of 80 degrees compared to the position before weir installation at 40, 50, and 60 degrees, respectively. The same condition was observed with a Froude number of 0.28. The difference between the dimensionless shear stress with the presence of a weir decreased an average of 1.2 times more than the situation without the presence of a weir, from the zero position to 70 degrees, with a Froude number of 0.28. The dimensionless shear stress significantly increased with a Froude number of 0.28 after weir installation, due to the appearance of two eddy vortices immediately after structure installation (Figure 12).

Table 4 lists the maximum dimensionless shear stress with and without a W−Weir. Bhuiyan (2009) investigated the effect of using a W−Weir on flow and sediment patterns in a meandering route [18]. The author discovered that the sediments deposited in the outer bends were generally larger and had a lower flatness coefficient after installing the W−Weir. We used these same positions that had higher dimensionless shear stress (Figure 12) in our study, resulting in the formation of a secondary flow in the outer bend. Bhuiyan discovered that the upstream sediment transfer in the center of the channel changes with a W−Weir; however, there are no significant changes to the maximum amount of sediment transfer, which could be a valuable consideration when deciding on riverbank protection plans.

**Table 4.** Maximum dimensionless shear stress with and without a W−Weir.

| Presence or Absence of a W−Weir | Froude Number | Maximum Dimensionless Shear Stress | Position (Degrees) |
|---|---|---|---|
| Without W−Weir presence | 0.17 | 1.8 | 20 |
| Without W−Weir presence | 0.28 | 5.78 | 40 cm after the bend |
| With W−Weir presence | 0.17 | 5.28 | 80 |
| With W−Weir presence | 0.28 | 9.74 | 80 |

The presence of a W−Weir causes homogeneous shear stress before the installation place of the weir. The maximum amount of dimensionless shearing stress with the presence of a W−Weir occurred in the position after the installation place, which was 5.28 for a 0.17 Froude number at 80 degrees. For the Froude number of 0.28 in 80-degree positions, the value reached 9.74. However, the maximum dimensionless shearing stress without the presence of a weir for the 0.17 Froude number in a 20-degree position is 1.8, and this value for the 0.28 Froude number becomes 5.28 in the 90-degree case.

The maximum dimensionless shear stress at the W−Weir position increased by 193.3 and 68.5 percent, respectively, with Froude number values of 0.17 and 0.28, compared to positions without a weir (Table 4). The maximum dimensionless shear stress increased by 221.1 percent with a Froude number of 0.28 in a location with a W−Weir, compared to a location without a W−Weir and a Froude number of 0.17. The maximum shear stress for both Froude numbers occurred in the region after the weir installation positions. This number was 84.4 percent higher with a Froude number of 0.28 than a Froude number of 0.17, indicating that an increase in the Froude number results in an increase in maximum shear stress.

Ghadoo and Shafai (2014) conducted research based on the effects of coarseness on the flow pattern in a location without a weir for a 90-degree bend, which agrees with our work. Ghadoo and Shafai (2014) performed studies using a coarseness of 5 mm, with a Froude number of 0.28, without the presence of a weir, and determined that the proportion of maximum shear stress occurred at the 20-degree position, also agreeing with our work [6]. The maximum dimensionless shear stress occurred in the presence of a W−Weir at the outer wall limits of 20 to 40 cm. Figure 12a,b indicate that a secondary flow forms in the outer wall after installing the W−Weir at the 90-degree mark (Table 4).

### 4. Conclusions

- The presence of a W−Weir stops outer bank erosion, but the bed topography does not change upstream of the W−Weir. Scour holes developed at the center of the flume downstream of the W−Weir due to the appearance of two eddy vortices downstream of the W−Weir. This scour hole was located far away from the banks and did not cause the bank to scour. This result matches with that of Abdollahpour, M., et al., who concluded that after installing a W−Weir, local scour increased and the bed upstream did not change appreciably, although their experiments were conducted in a mild bend.

- We tested three locations (30-, 60-, and 90-degrees) with two different Froude numbers for this study, and established that a lower scour depth and volume of sediment removal occurred when the weir was located at the end of the bend. This study corroborated Ghadoo and Shafai's finding that suggested that 70 degrees of the bend was the ideal location to install a W−Weir. Bhuiyan et al. also determined that immediately downstream of the curve, where the flow pattern has severe turbulence, was the best place to install the W−Weir.

- The results of this experimental study indicate that variations in structure installation angle and Froude number affect the scour characteristics downstream of a W−Weir. For instance, the erosion on the outer bend was reduced by 39 and 37 percent, respectively, with a Froude number of 0.28 and W−Weir installation at a 90-degree angle, compared to those installed at 30- and 60 degrees.

- The external wall range had the greatest dimensionless shear stress, at 20 to 40 cm. In addition, a secondary flow was observed in the external wall at a 90-degree angle after the W−Weir installation, in flow patterns. The maximum depth of bed erosion occurred in the same situation, which shows that the results are converging.

- Future studies are proposed to investigate the performance of the W−Weir in the bed with non-uniform materials, the effect of using two or more W−Weirs at once, and the effect of using different W−Weir sizes in the 90-degree bend.

**Author Contributions:** Conceptualization, V.A.; methodology, V.A.; software, V.A.; validation, V.A., M.S.B. and Y.H.L.; formal analysis, V.A.; investigation, V.A.; resources, V.A.; data curation, V.A.; writing—original draft preparation, V.A.; writing—review and editing, V.A., M.S.B. and Y.H.L.; visualization, V.A.; supervision, M.S.B.; project administration, M.S.B.; funding acquisition, M.S.B. All authors have read and agreed to the published version of the manuscript.

**Funding:** This research received no external funding.

**Institutional Review Board Statement:** Not applicable.

**Informed Consent Statement:** Not applicable.

**Data Availability Statement:** Not applicable.

**Acknowledgments:** We thank Ideh Golrokh for collecting some of the data and laboratory analysis.

**Conflicts of Interest:** The authors declare no conflict of interest. The funders had no role in the design of the study; in the collection, analyses, or interpretation of data; in the writing of the manuscript; or in the decision to publish the results.

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
