# Peer review of "Flow Pattern and Erosion in a 90-Degrees Sharp Bend around a W−Weir"

_water, doi:10.3390/w15010011_

Round 1
Reviewer 1 Report
The paper presents results of an experimental study of the best position of a W-weir in a movable bed flume.
- There is a similar paper reference of the same authors:
Experimental study of the effect of W-weir on reduction of scour depth at 90 degree sharp bend.
Author(s) : Atashi, V. ; Shafai-Bajestan, M. ; Golrokh, I.
Author Affiliation : Department of Hydraulic Structures, Faculty of Agriculture, Shahid Chamran University, Ahwaz, Iran.
Author Email : vida.atashi@yahoo.com
Journal article : Journal of Water and Soil 2016 Vol.30 No.2 pp.Pe392-Pe404 ref.16
The authors and the editor should consider if this work is different from the one already published or the author should explain what are the main improvements/differences.
- Materials and methods paper section should include a characterization of the sediments used in the experiments. And morphological results should be discussed accordingly. Eventual limitations resulting from the adopted bed sediments sizes and densities, on the different analyzed experiments results should be presented.
Other comments:
Line 21: Complete/Rephrase “… power without a weir is more than …” eg. … power without a weir is more intense than …
Line 24: Change “Fraud” to Froude
Line 66: Remove the final mark “respectively. [5].”
Line 77: Add a final mark.
Line 112: Correct “Knowing precisely how the to scour morphology …”
Figure 1: Is “y” in Figure 1 a) the same as represented in Figure 1 b? If not use a different variable.
Line 156: Is this a section title? Remove or number it.
Figure 2: Indicate the units of the numbers. Complete the figure caption (eg. Figure 2. Plan view of laboratory flume and the test bend (all dimensions in meters)
Line 184: Change “as follows: The flow slowly” to “as follows: the flow slowly”
Line 190: Use standard units along the paper. In this case “L/s”
Lines 216-221: The aims of the paper were already presented at the introduction paper section.
Figures 4 and 5: Insert the flow direction
Lines 253-254: Repetition. Already mentioned at paper line 228
Lines 279-280: Remove the numbers at the beginning
Lines 306-307: The authors should further explain and justify why 70 degrees was chosen for the positon of the weir, since their work did not include this position.
Lines 323-327: Repetition of the contents of lines 315 -321
Line 341: The authors decided to analyze a 70 degrees’ installation position. Why are they presenting results for a 45 degrees’ position installation?
Lines 394-402: This contents are more appropriate for the materials and methods paper section.
Lines 422-423 – The title of the Table 2 is not in the correct position.
Lines 435-458 – Same comment as for lines 394-402
Figure 12 – The representation of the bend is different from previous Figures. Insert the angle in order to allow an easy comparison.
Line 459: Change “Figures” to Figure
Line 460: Change “bed in both” to bed with both
Lines 472-473: The Froude number Is not the most appropriate non-dimensional hydraulic number to relate turbulence with inertial forces of the flow. Authors should further justify explain their option.
Line 505: Check the position of the table title
Lines 507-513: Consider rephrase in a clearer way.
Reviewer 2 Report
The manuscript needs a large work of revision (see my comments in the attached marked manuscript).
A very careful proof-reading is absolutely needed to improve the quality of the English.

Round 2
Reviewer 1 Report
The authors followed all the suggestions of the first revision.
Author Response
Thank you.
Spell checks have been done.
